# Fragment-Based Ab Initio Molecular Dynamics Simulation for Combustion

**DOI:** 10.3390/molecules26113120

**Published:** 2021-05-23

**Authors:** Liqun Cao, Jinzhe Zeng, Mingyuan Xu, Chih-Hao Chin, Tong Zhu, John Z. H. Zhang

**Affiliations:** 1School of Chemistry and Molecular Engineering, Shanghai Engineering Research Center of Molecular Therapeutics & New Drug Development, East China Normal University, Shanghai 200062, China; caoliqun1994@163.com (L.C.); mingyuan_xu2014@163.com (M.X.); zhjin@chem.ecnu.edu.cn (C.-H.C.); 2Department of Chemistry and Chemical Biology, Rutgers University, Piscataway, NJ 08854, USA; jinzhe.zeng@rutgers.edu; 3NYU-ECNU Center for Computational Chemistry, NYU Shanghai, Shanghai 200062, China; 4Department of Chemistry, New York University, New York, NY 10003, USA

**Keywords:** FB-AIMD, molecular dynamics, Jacobi coordinate, methane combustion, reaction mechanism

## Abstract

We develop a fragment-based ab initio molecular dynamics (FB-AIMD) method for efficient dynamics simulation of the combustion process. In this method, the intermolecular interactions are treated by a fragment-based many-body expansion in which three- or higher body interactions are neglected, while two-body interactions are computed if the distance between the two fragments is smaller than a cutoff value. The accuracy of the method was verified by comparing FB-AIMD calculated energies and atomic forces of several different systems with those obtained by standard full system quantum calculations. The computational cost of the FB-AIMD method scales linearly with the size of the system, and the calculation is easily parallelizable. The method is applied to methane combustion as a benchmark. Detailed reaction network of methane reaction is analyzed, and important reaction species are tracked in real time. The current result of methane simulation is in excellent agreement with known experimental findings and with prior theoretical studies.

## 1. Introduction

Combustion is the earliest form of chemical reaction consciously used for heating and cooking by human beings for thousands of years. With the depletion of fossil energy and extremely serious environmental problems, the need for a thorough understanding of combustion is more urgent than ever. Modern combustion usually occurs at high temperature and pressure, which makes the experimental study relatively difficult. In the past few decades, theoretical methods have gradually become one of the main tools for studying combustion reactions. A breakthrough in this area is the development of reactive force field (ReaxFF) by van Duin et al. [1,2,3]. By combining with classical molecular dynamics (MD) engine, ReaxFF allows one to simulate the dynamics of breaking and formation of chemical bonds and construct the interwoven reaction networks, often from a single MD trajectory. Benefiting from its efficiency, ReaxFF has been used extensively to investigate the reaction mechanism of many complex systems including combustion [4] and pyrolysis, etc. [5]. Improved ReaxFFs have also been reported and applied to improve the accuracy [6].

However, as an empirical model, the accuracy and reliability of the reactive force field remain a major concern. In comparison, the quantum mechanical (QM) method is more rigorous and accurate and has been widely used to study the mechanism of predefined elementary reactions [7,8], such as the calculation of reaction barriers and reaction rates. MD simulation, in which the potential energies and atomic forces are calculated by QM methods, is known as ab initio MD simulation (AIMD) [9,10,11,12,13,14,15]. However, although being more accurate than force fields, it is currently impractical to use AIMD to simulate complex reaction systems such as combustions due to exorbitant computational costs of QM calculations.

In the past decades, considerable efforts have been made to extend the applicability of QM calculation to large chemical systems. Among these efforts, the fragmentation-based QM methods have made significant advances [16,17,18,19,20,21,22,23,24,25,26,27,28,29,30,31]. In a fragment quantum approach, a given large system is divided into fragments whose properties are calculated separately by quantum mechanics, and the properties of the whole system can be obtained by taking proper linear combinations of the properties of the individual fragments. The main advantage of fragment-based QM methods lies in its linear scaling, trivial parallelization, and easy employment of high-level ab initio methods, since the size of each fragment is small. As combustion reactions are dominated by short-range collisions, the fragment-based QM method should be suitable for the dynamics simulation of combustions. Some fragment or analogous methods have already been used to calculate quantum mechanical energies for complex chemical reactions [32,33].

In our previous studies, we combined fragment methods with the MD engine and performed AIMD simulations for large molecular systems including protein, RNA, and hydration of metal ions [20,34,35]. In this study, a fragment-based AIMD (FB-AIMD) for combustion is developed, and its performance on the calculation of potential energies, atomic forces, and AIMD simulation of methane combustion are systematically investigated.

## 2. Results and Discussion

### 2.1. The Accuracy and Efficiency of the FB-AIMD Method

Four different systems were chosen to test and verify this boundary condition method, including methane combustion (18 CH_4_ + 36 O_2_), ethane combustion (7 C_2_H_6_ + 28 O_2_), cyclooctane combustion (2 C_8_H_16_ + 24 O_2_), and dodecane pyrolysis (3 C_12_H_26_). To obtain more diverse structures, these four systems were simulated with the ReaxFF under the 3000 K for 250 ps employing the NVT canonical ensemble, respectively. The structures were randomly selected from the trajectories in these four systems to be calculated by the FB-AIMD method. These systems consider the diversity of different types of hydrocarbon fuels and are more reliable in the verification of the method.

The choice of the threshold distance λ is important, because it determines whether the interaction between the two monomers is considered or not. Therefore, we first checked the influence of λ on the convergence of the calculated potential energy. Figure 1 shows the calculated potential energies of four systems as a function of λ value, with results of the full-system QM calculation used as reference. As can be seen, the potential energies of all systems are close to convergence at λ = 3.5 Å, which is thus adopted as the cutoff distance. The structures of these four systems are shown in Figure 1, which are taken from the MD simulation trajectories randomly.

We also need to check the accuracy of the FB-AIMD method for interaction energies as compared to the exact full system quantum calculation. Table 1 shows the comparison of the potential energies calculated by FB-AIMD with that of full-system QM calculation. The differences between these two methods are all within 4 kcal/mol for the four systems, as shown in Table 1. This clearly verified the accuracy of the FB-AIMD method. Similar to the interaction energy, atomic forces are also calculated by the many-body expansion. As shown in Figure 2, the atomic forces calculated by the FB-AIMD method agree very well with the atomic forces from the full-system QM calculation. Although the values of the forces can be as large as ±200 kcal/(mol·Å), the root mean square errors (RMSEs) between the results calculated by FB-AIMD and full-system QM methods do not exceed 0.5 kcal/(mol·Å) as shown in Figure 2.

Figure 3 shows the comparison of efficiency between FB-AIMD and full-system QM calculation. In order to verify the relationship between calculation efficiency and system size, these systems with different numbers of atoms are used for testing. The number of atoms in these systems are 9, 54, 108, 162, 216, 270, 324, 342, 360, 405, 432, and 450, respectively, because we want to keep the ratio of the number of oxygen to that of methane in each system to 2:1. It can be seen that, with the increase of the number of atoms in the system, the CPU time of the full-system QM calculation increases cubically, while the CPU time of the FB-AIMD method increases almost linearly. With the increase of the system size, the advantages of the fragmentation method becomes more evident. The FB-AIMD method greatly saves the computational cost so that it is possible to simulate the process of combustion with a longer simulation time and a larger system size.

### 2.2. Molecular Dynamics Simulation of Methane Combustion

To benchmark the performance of FB-AIMD, the combustion of methane was used in our study, as it has been extensively studied, and abundant experimental/computational results are accessible [36,37,38,39,40]. A cubic box containing 18 CH_4_ and 36 O_2_ molecules was constructed with a density of 0.35 g/cm^3^ (Appendix A). The system was firstly preheated to 4000 K with the ReaxFF under the NVT canonical ensemble for 10 ps. It is reported that ReaxFF is a reliable method when it is applied in methane combustion, and the discovered species can also be found in experiments [40]. Then, the final snapshot of the preheated process was used to simulate with FB-AIMD method for the other 10 ps, and the time step was set to 0.2 fs. Thus, the initial state of the FB-AIMD simulation already contains some free radicals and intermediates. To verify the energy conservation of the FB-AIMD method, we performed a MD simulation of 400 steps under the NVE ensemble, starting from a random snapshot taken from the AIMD simulation of methane combustion with the total energy of this snapshot as the reference energy. As can be seen in Appendix A, the total energy is conserved with a deviation less than 2 kcal/mol.

The changes of main reactants and products during the FB-AIMD simulation are plotted in Figure 4. During the first 3 ps, the number of methane was reduced from 8 to 1, while the number of oxygen molecules was reduced from 32 to 15. Due to the high temperature, the reactants are consumed quickly. Besides, the number of CO was increased from 0 to around 10. It was followed by a gradual increase of CO_2_, and by 4 ps, the methane was almost completely consumed. The number of H_2_O is relatively small during the simulation. Some important free radicals were found during the MD simulation, including ·H, ·CH_3_, ·HO_2_, ·OH, which greatly affects the combustion process [41,42]. The variation of the numbers of these free radicals with the simulation time is shown in Figure 5. Many hydrogen radicals were produced during the FB-AIMD simulation. As shown in Figure 5, the number of methyl radicals decreases at the beginning, while the number of ·CHO increases after 1 ps. During the 10 ps FB-AIMD simulation, the variation of the number of ·HO_2_ radicals fluctuated, showing that it is actively involved in the reactions. The number of hydroxyl radicals increases obviously after 2 ps, which decreases after 9 ps. Formaldehyde appears as a major intermediate in the reaction pathway, and its population is shown as a function of time together with the other radicals in Figure 5.

The trajectory was further analyzed by the ReacNetGenerator software [43], which can construct reaction networks automatically from atomic coordinates. The main reaction network detected from the trajectory is shown in Figure 6. The passage of reactions is numbered. Due to the high temperature, most of the reactions during methane combustion are reversible. In step I, the oxidation of methane started with the abstraction of the H from CH_4_ by O_2_ to form methyl radicals and ·HO_2_, and the other way is through C-H bond cleavage to produce methyl and ·H, which can be found in the experiment. After this step, the methyl can combine an O·· free radicals or O_2_ to form ·CH_3_O radicals or CH_3_O_2_ molecules, and the reaction from ·CH_3_ to ·CH_3_O is also involved in GRI-Mech 3.0. The reactions of C-H bond scission also occur with methyl to form ·CH_2_, which is not mentioned in previous work. It may be due to the high temperature that caused the reaction to take place. Then, the ·CH_3_O radicals release a ·H to form formaldehyde (HCHO), in step III, and the HCHO also can be produced by the combination of ·CH_2_ and O··. The former reaction can be found in the mechanisms provided by GRI-Mech 3.0 [39]. In the next step, the formaldehyde continues to combine with ·H, ·OH, or O_2_ to form ·CHO radicals, which can be found in previous work [3]. Then the ·CHO radicals are consumed to generate CO molecules or ·CO_2_H radicals through combining with ·OH or O·· radicals in step V. In the final step, the production of CO_2_ mainly comes from the conversion of CO, and a part of CO_2_ is generated from ·CO_2_H losing a ·H, which are found in experimental and theoretical studies. The typical snapshots from the initial state to the final product and some important intermediates that can be observed during the simulation intuitively are shown in Appendix A. There are lots of reactions of hydroxide in the simulation trajectories, and it will not be introduced in detail in this work.

## 3. Theoretical Methods

The workflow of the FB-AIMD method is shown in Figure 7. In this approach, the potential energy and the atomic forces are calculated by the fragment method, while the dynamics of the nucleus are driven by molecular dynamics.

For a given structure from step *i* of simulation, the Open Babel [44] package is firstly employed to identify the connectivity between atoms based on atomic Cartesian coordinates. Depth-first search [45], an algorithm designed for traversing graph data structures, is then used to detect molecules (including radicals) based on the connectivity of atoms. The energy and atomic forces of this snapshot are calculated by the FB-AIMD method. When the step updates to *i + 1*, the structure of the system will change according to the atomic forces in the step *i*, and the atoms will be recombined into new fragments. According to the many-body expansion [46,47], the potential energy of a system that contains *N* molecules can be written as:(1)$$V=V_{1}+V_{2}+V_{3}+\cdots +V_{N}$$
where $V_{N}$ (with *N* > 1) represents *N*-body interactions. The one-body energy is the sum of potential energies ($E_{i}$) of all monomer:(2)$$V_{1}=\overset{N}{\underset{i}{\sum }}E_{i}$$
and the two-body interaction is defined by
(3)$$V_{2}=\underset{i>j}{\sum }E_{ij}-E_{i}-E_{j}$$
where $E_{ij}$ is the energy of the dimer consisting of two monomers *i* and *j*.

In the current FB-AIMD implementation, Equation (1) is truncated at the two-body term to balance the computational cost and accuracy. The two-body cutoff is reasonable for combustion reactions, since three-body reactions are rare. For each monomer and dimer, the potential energy and atomic forces are calculated at the MN15/6-31G(d) level by using the Gaussian16 program [48]. The MN15 is a Kohn–Sham global-hybrid exchange-correlation density functional with broad accuracy for multireference and single-reference systems and noncovalent interactions, in which bond energies, reaction barrier heights, and hydrocarbon thermochemistry were included in its training sets [49]. Larger basis sets like 6-31G(d,p) and 6-31++G(d,p) have been tested, and their effect on computed energies are minimal, as shown in Appendix A. In addition, to consider the spin-polarization of multiradicals, the initial wave function of the dimer is constructed by the combination of the wave function of each monomer. In our calculations, free radicals were considered, but ions were not. This is because the breaking of most of the chemical bonds in the combustion reaction of hydrocarbon fuels will only produce free radicals instead of ions.

In the energy calculation of this FB-AIMD method, if the closest distance between two monomers *i* and *j* is less than or equal to a threshold distance λ, the interaction between these two monomers is then calculated by quantum mechanics. The energy and atomic forces are calculated by the FB-AIMD method, which is passed to the MD engine in the LAMMPS program [50].

The combustion reaction is mainly driven by short range intermolecular collisions; the influence of long-range interactions is negligible. Thus, the combustion reactions are not sensitive to the size of the system, and periodic boundaries are not necessary. The hard wall boundary condition was used in this work to keep total energy conserved. To avoid unphysical collisions on the boundary, we employed the Jacobi coordinate [51]. In the MD simulation, when a molecule hits the boundary, the velocity of its center-of-mass, instead of individual atoms, will be reversed, which can avoid unphysical bond scratches and additional work caused by the collision. The detailed relationships between the Jacobi coordinates and Cartesian coordinates are given in the Appendix A.

## 4. Conclusions

In this work, the FB-AIMD method was developed to perform AIMD simulation for combustion in which the interaction energies and atomic forces are calculated by a fragment approach. In this fragment method, the interaction energy between any two monomers within a distance of 3.5 Å is calculated by the quantum method, and three-body or higher multibody interactions were neglected. The calculated interaction energies and atomic forces by the FB-AIMD method are in excellent agreement with those from the full-system calculations. More importantly, the computational cost of the FB-AIMD method increases almost linearly with the increase of the system size, and it is more suitable for highly parallel computation. This FB-AIMD method was applied to methane combustion, and the Jacobi coordinate boundary condition was employed to avoid possible unphysical effects at the boundary. The detailed reaction mechanism was extracted from the trajectory, and the main reaction paths obtained agree well with experiments and previous theoretical studies. Further improvement of the FB-AIMD method will focus on two aspects: the first one is to consider the long-range interaction, and the second one is to further reduce its computational cost, such as using deep learning methods to learn the potential energy surface and atomic force of the existing trajectory on-the-fly and minimize the QM calculations.

## Figures and Tables

**Figure 1 molecules-26-03120-f001:**
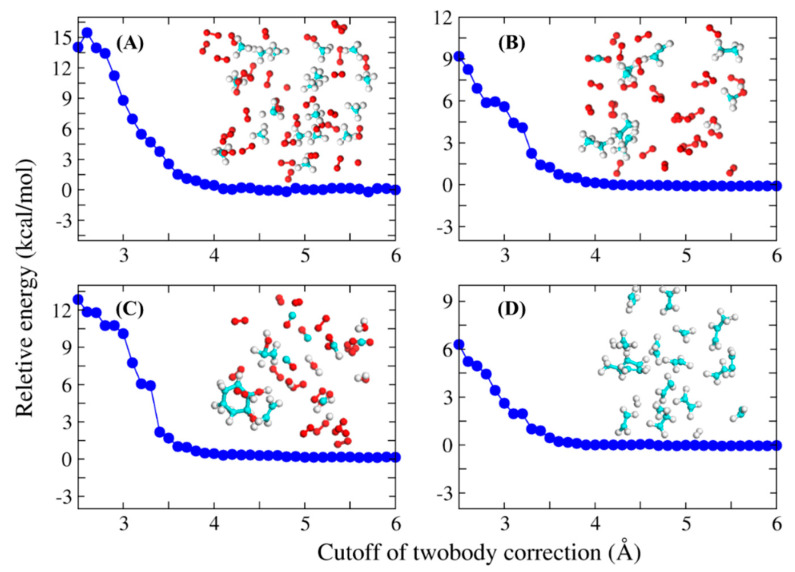
Relative potential energies (in kcal/mol) calculated by the FB-AIMD method are shown as a function of the distance threshold of two-body interaction for four different simulation systems, and the structures of these four systems were inserted in the figure. (**A**) random snapshot taken from the methane combustion system. (**B**) random snapshot taken from the ethane combustion system. (**C**) random snapshot taken from the cyclooctane combustion system. (**D**) random snapshot taken from the dodecane pyrolysis system. Potential energies calculated by the full-system QM methods are taken as reference. All QM calculations were performed at the MN15/6-31G(d) level.

**Figure 2 molecules-26-03120-f002:**
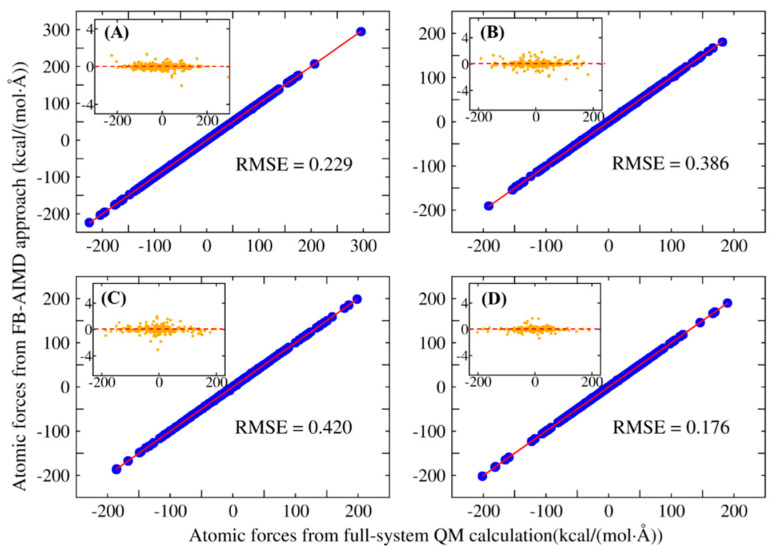
Correlations between atomic forces calculated by full-system QM calculation and by the FB-AIMD method for structures taken from the trajectories of the four different systems. (**A**) random snapshot taken from the methane combustion system. (**B**) random snapshot taken from the ethane combustion system. (**C**) random snapshot taken from the cyclooctane combustion system. (**D**) random snapshot taken from the dodecane pyrolysis system. A separate panel on each panel indicates the difference between the two values on the y axis. All QM calculations were performed at the MN15/6-31G(d) level. The unit of RMSE is kcal/(mol·Å).

**Figure 3 molecules-26-03120-f003:**
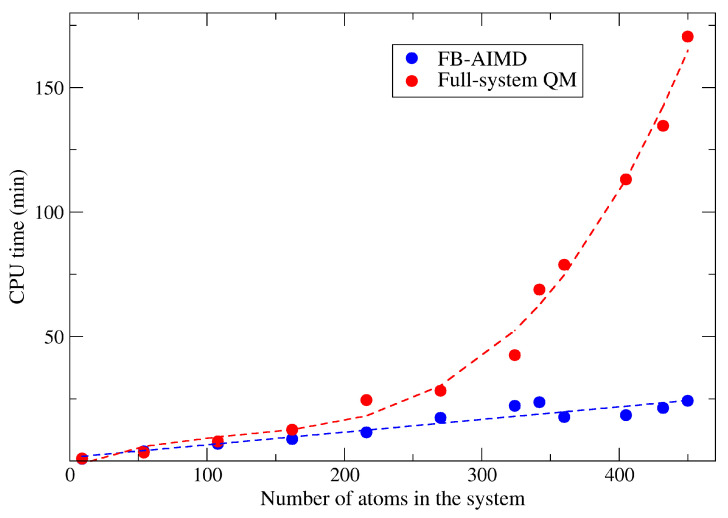
Comparison of the computational efficiency of the FB-AIMD and full-system QM methods on a single Linux workstation with 28 CPU cores. All QM calculations were performed at the MN15/6-31G (d) level.

**Figure 4 molecules-26-03120-f004:**
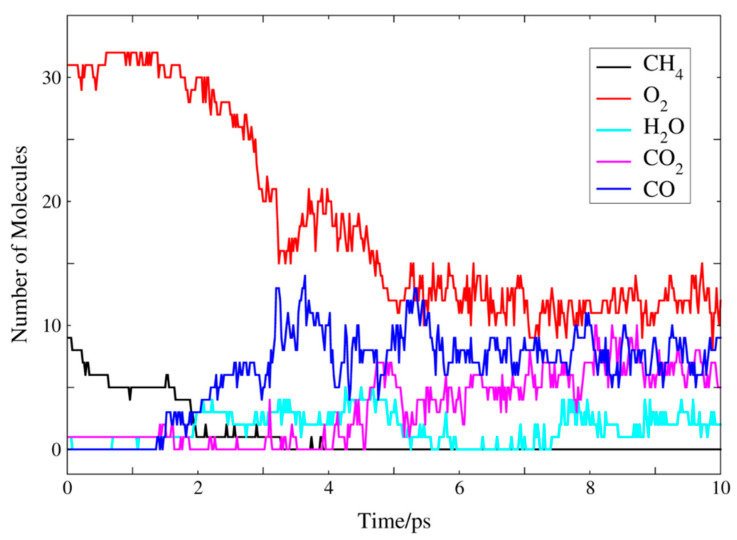
The numbers of main reactants and products as functions of the simulation time.

**Figure 5 molecules-26-03120-f005:**
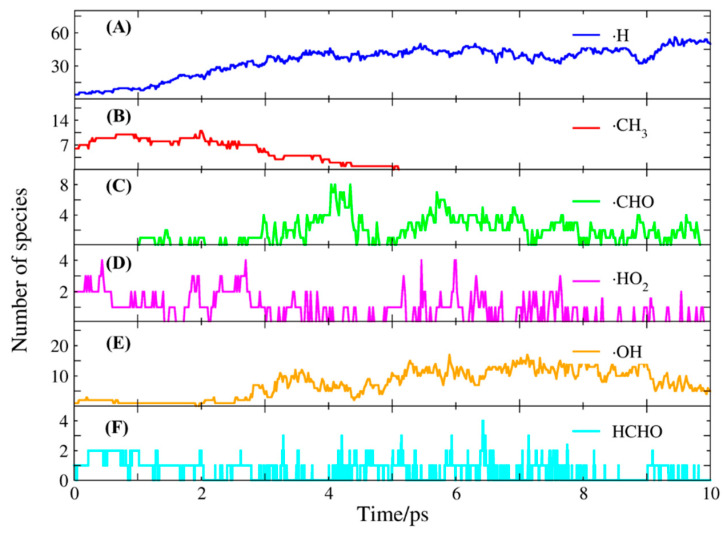
The numbers of species as functions of the simulation time. (**A**) the evolution of ·H radicals. (**B**) the evolution of ·CH_3_ radicals. (**C**) the evolution of ·CHO radicals. (**D**) the evolution of ·HO_2_ radicals. (**E**) the evolution of ·OH radicals. (**F**) the evolution of HCHO molecules.

**Figure 6 molecules-26-03120-f006:**
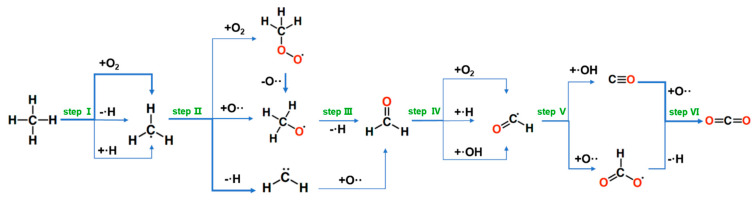
One of the reaction paths detected from the trajectories. The width of an arrow is proportional to the number of reactions. The radical is denoted by a dot in front of the atoms (molecules).

**Figure 7 molecules-26-03120-f007:**
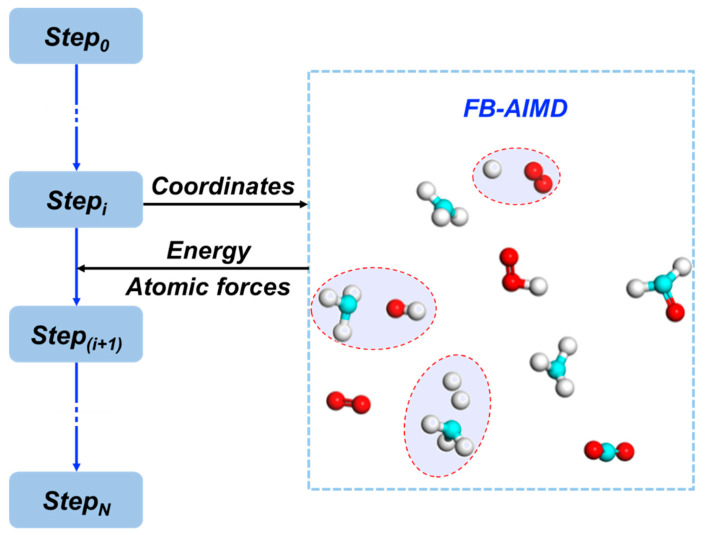
Fragmentation scheme of the FB-AIMD approach. The shadow circles indicate that two neighboring molecules can be treated as a dimer when their distance is smaller than a cutoff value (3.5 Å). Here the hydrogen, carbon, and oxygen atoms are colored in white, cyan, and red, respectively.

**Table 1 molecules-26-03120-t001:** Potential energies of the structures taken from the trajectories of the four different systems, calculated by the full-system QM calculations and FB-AIMD, respectively. All calculations were performed at the MN15/6-31G(d) level.

System	Number of Atoms	Full-System (au)	FB-AIMD (au)	Energy Deviation (kcal/mol)
System Ⅰ	162	−6132.09287	−6132.08737	3.452
System Ⅱ	112	−4762.42315	−4762.41884	2.703
System Ⅲ	96	−4232.35592	−4232.35346	1.541
System Ⅳ	114	−1415.00577	−1415.00222	2.229

## Data Availability

Data are available from the authors on request.

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
