# Peer review of "Fragment-Based Ab Initio Molecular Dynamics Simulation for Combustion"

_molecules, 2021, doi:10.3390/molecules26113120_

Round 1

Reviewer 1 Report

The manuscript "Fragment-based Ab Initio Molecular Dynamics Simulation for 2 Combustion" by Cao et al. presents a specific way to use fragmentation to approximate the evaluation of ab initio interactions for the purpose of ab initio molecular dynamics, specifically the simulate reactive processes during combustion. While the work shows potential, the manuscript is in need of a major revision before it is suitable for publishing.

The manuscript is written and presented clearly, with only very minor editing needed, such as "oftern" on line 32 and a couple of plurals or articles to adjust. Both the abstract and the introduction are clear and well-suited for the purpose.

I think the explanation of the methodology needs to be improved.  My biggest concern lies in the details of the fragmentation procedure, which is explained only as depth-first search taken from the Open Babel package. However, as the authors aim to describe reactivity, they should explain in detail how they deal with the change of the fragmentation during the dynamics of the system and maintaining the resulting potential energy surface smooth. Showing explicitly that the PES remains smooth as the fragmentation changes in time would be very helpful in convincing readers, i.e., potential users of the method, that the method is useful and can be trusted.

They mention that radicals are also described, but again, how this is done and how consistency is maintained globally in the whole system should be explained. How are charges of the fragments assigned? Can the method distinguish between splitting into radicals vs. ions, for example?

As a minor point, I think the outline in Figure 1 is potentially misleading. The forces are calculated for the geometries at step i and immediately used to propagate further to step i+1. The arrow to step i+1 is therefore confusing. In my opinion, the left part is not needed at all, considering that it is standard MD and the methodology of this manuscript only pertains to how to evaluate the interactions faster.

The ab initio calculations themselves run in open boundary conditions, while dynamics is confined by reflective walls. Given the small size of the systems, less than 100 heavy atoms, the surface of this system is substantial relative to its total size, which can impact the result of the ab initio calculation, be in full or in fragments. Periodic boundary conditions would help avoid that. Have the authors considered such a case? Can their method be used in periodic boundary conditions? Would it be preferable to reflective boundaries?

I would like to suggest an improvement in the clarity of the tests in Figures 2 and 3. In Figure 2, showing that relative energies between different geometries (rather than the absolute energy of each geometry separately) converge would be good. In Figure 3, the combination of the number and size of the points means that we do not really see what the deviations are. It would help, as is customary in these comparisons, to add to each panel a separate panel with the difference between the two values on the y axis. These changes are not needed but would, in my opinion, help "sell" the accuracy of the method.

Honestly, I find the computational cost argument a little unconvincing. There are no specifics of the full calculation or any information on if/how computational cost was optimized to make it a meaningful reference. As very rough context (but the authors could get specific reference information themselves), CP2K will evaluate a hybrid functional for a "standard" box of 64 water molecules in periodic boundary conditions with a TZV2P basis set most likely under 2 minutes on the mentioned 28 core computational node. Perhaps the authors want to focus more on the scaling and performance for larger systems and/or the ability to reach longer time scales. The system that they show in detail has 90 heavy atoms and 162 atoms in total, and they run it for a total of 20 ps. That would be very doable even with the full calculation, no fragmentation. Plus, at this system size, even their own comparison shows very little difference in the computational cost of the full vs. fragmented calculation.

The employed basis set, 6-31G(d) is on the smaller side. Did the authors test larger basis sets?

Finally, the Conclusions are rather brief, I would expect some outlook to be included. What new calculations will be possible thanks to the proposed methodology? What are the remaining open questions and possible directions to improve the method?

Reviewer 2 Report

Reviewer’s comments on draft 1171076

In this paper, Authors describe the development of a fragment-based ab initio molecular dynamics method to simulate high temperature combustion reactions. The many-body potential energy expression is truncated at the two-body term to save computational costs. The two-body interactions are computed only if the distance between two fragments is within a given threshold value. The proposed method scales linearly with the system size and, as stated by Authors, should be easily parallelizable.

Four systems were considered as test cases and the relative accuracy was verified respect to the full quantum approach.

I think that the subject of this study is somehow interesting, but the paper is not very easy to read as it lacks clearness and a good results presentation. The bibliography is rather limited, 39 citations for a ongoing developing topic are not to many. Finally, the authors stated excellent agreement with experimental and theoretical results, although no clear comparison is reported in the text.

To this end, many necessary and important improvements must be introduced before the publication in the Molecules journal. The present work cannot be accepted in the present form and I recommend Major Revision.

Here my comments in details:

Comment 1:

About the energy calculation in the proposed FB-AIMD method, how is the threshold distance between monomers evaluated? Taking into account the density of the simulated system, which dimer results if two species are close to a third one? The Authors should be clearer on this point.

In the Theoretical method section, at line 105, Authors state that the first system under study is composed of 12CH4+24O2, for a total of 108 atoms. In table 1 second column, 162 atoms are reported instead. Please clarify and correct in the text.

Comment 2:

How does the proposed fragment method here compare with previous well-established ones (i.e. in Ref. #11) in terms of accuracy and costs? Is this method particularly well implemented, or any method truncating the potential evaluation at the two-body term will scale linearly? What is new here? Please explain.

Comment 3:

About the accuracy of the method. Authors compare their FB with DFT results on relative energies. Have Authors checked energy barriers with their method vs DFT? If bond breaking and formation reactions involve radical species, how has this potential been checked, since also DFT can suffer from the inherently multi refence character of the wavefunction? Please show some energy scans for bond breaking reaction and compare this result with experiments.

Comment 4:

Which snapshots are shown in figure 2? The Authors should expand the caption of figure 2 reporting a description for each labeled panel.

In figure 3 on the y-axis an MFCC approach is invoked but it is never stated in the text. Authors should clarify this point. Furthermore, the caption should be improved explaining what is reported in each labeled panel.

In figure 4, the legend reports the MFCC approach, as for the previous point, Authors should explain what they refer to.

Comment 5:

Considering the breaking and formation of chemical bonds during the simulations, how the basis set was chosen? Would not a larger basis set with diffuse functions on carbons and hydrogens be more suitable?

Figure 4 shows the efficiency comparison between FB-AIMD and full quantum approach. The figure shows that the total number of atoms is greater than 400. In the text, the largest system is made up of 114 atoms. Authors should add some details about it.

Comment 6:

In the 3.2 paragraph, Authors should quantify the terms "sharply", "rapidly" at lines 163-164. Please rephrase also the sentence at line 175 "The number of hydroxyl radicals increases obviously after 2ps and they were consumed after 9ps."

Comment 7:

In the Abstract (lines 20-21) Authors declare an excellent agreement with experimental and theoretical studies, but these agreements cannot be inspected in any part of the text. Please expand the Results, Discussions and also Conclusions by explicitly stating what is in agreement and what is not.

Comment 8:

The discussion about Figure 7 should be expanded and made clearer. Numbering each passage discussed might help the reader.

Comment 9:

The methane/O2 fragment-QM ab initio dynamics is started after a ReaxFF dynamics at 4000K. Since “as an empirical model, the accuracy and reliability of the reactive force field remain a major concern” (pg.1), can the intermediates and radicals distribution at the beginning of the ab initio dynamics (so generated by the previous reactive FF dynamics) be considered correct? Please clarify.

Comment 10:

In pg.3, “the potential energy and atomic forces are calculated at the MN15/6-31G(d) level by using the Gaussian16 program” is stated and then “in this FB-AIMD method, the energy and atomic forces are calculated by the LAMMPS program”. LAMMPS is a classical Molecular Dynamics code. Has it been used to integrate the equations of motion, taking the ab initio forces from Gaussian? Please re-formulate these sentences.

Comment 11:

In pg.7:

  1. “·CH3O radicals combine with ·H to form formaldehyde (HCHO)”. In fig.7, a -·H symbol seems instead an homolytic bond cleavage.
  2. Formaldehyde appears as a major intermediate in the reaction pathway. Despite not being a radical, maybe its population could be shown as a function of time together with the other (radical) intermediates, figure 6.
  3. “Formaldehyde … which can lose its H atoms”. A +·H symbol in fig.7 suggests instead a reaction with a hydrogen radical.
  4. “The production of CO2 mainly comes from the conversion of ·CO2H”. According to the distinct arrows thicknesses, it seems that CO2 is mainly produced from CO + O··.

Minor points:

What does it mean the term "conventional" relative to the full QM approach?

Please correct dynamic as dynamics in the Abstract, line 11.

Please correct oftern as often at line 32 in the Introduction

Please correct “a distance of 3.5Å distance”, line 204

Please correct Benefitting as Benefiting at line 33 in the Introduction

Please add the following citation at line 42 in the Introduction when AIMD is mentioned:

  • B. Schlegel, J. M. Millam, S. S. Iyengar, G. A. Voth, A. D. Daniels, G. E. Scuseria and M. J. Frisch, J. Chem. Phys., 2001,114, 9758–9763.
  • B. Schlegel, S. S. Iyengar, X. Li, J. M. Millam, G. A. Voth, G. E. Scuseria and M. J. Frisch, J. Phys. Chem., 2002, 117, 8694–8704.
  • S. Iyengar, H. B. Schlegel, G. A. Voth, J. M. Millam, G. E. Scuseria and M. J. Frisch, Israel J. Chem., 2002, 42, 191–202.
  • B. Schlegel, Bull. Korean Chem. Soc., 2003, 24, 837–842.
  • Rega, S. S. Iyengar, G. A. Voth, H. B. Schlegel, T. Vreven and M. J. Frisch, J. Phys. Chem. B, 2004, 108, 4210–4220.

Please rephrase the caption of Figs. 5 and 6

Reviewer 3 Report

A fragment based ab inito molecular dynamics (FB-AIMD) method is proposed in this article to tackle to computational costs involved in the combustion processes. Molecular interactions up to two-body with a cutoff value of 3.5 Å are considered. Methane, ethane, cyclooctane combustion reactions and dodecane pyrolysis were taken as examples, and the energy deviations compared with the full ab initio simulations are under 4 kcal/mol. While the overall article is well-written, some misprints and spelling mistakes need to be corrected before it can be published. They are mentioned here below:

Spelling mistakes/Misprints

Line 32: Oftern --> often

Line 69: I think the cutoff value can be mentioned here in parenthesis.

Please use either mn15/6-31g(d)  (Table 1) throughout or MN15/6-31G(d) through out – Line 87.

At many places there is no single space between temperature and its unit (for example 3000K at Line 108), and time and its units (for example 250ps at Line 108). These things require a space between like 3000 K, 250 ps, etc.,

Line 127 and 131; Fig. 3 to Figure 3.

Line 142. kcal/(mol·Å).The structures … --> kcal/(mol·Å). The structures …

Line 145: I am just wondering whether the CPU time is increasing cubically or exponentially?

Figure 4. y-axis label CPU time(min) --> CPU time (min)

Lines 154-156: Please give some appropriate references here for the combustion of methane.

Line 162: Fig. 5 --> Figure 5. I think throughout the article, the convention should be uniform.

Line 204: 3.5Å  --> 3.5 Å

Round 2

Reviewer 2 Report

I think Authors have addressed my comments/revisions and I thank them for considering my opinions. The paper is now definitively improved, and I feel now confident recommending this work for publication in Molecules journal in the current form.

Author Response

Thanks for your comments.  We corrected some errors in  English in the revised version.